# An Efficient CS-Based Spectral Peak Search Method

**DOI:** 10.3390/s22187025

**Published:** 2022-09-16

**Authors:** Bingbing Chen, Yufa Sun

**Affiliations:** 1School of Electronic and Information Engineering, Anhui University, Hefei 230061, China; 2Anhui Province Key Laboratory of Simulation and Design for Electronic Information System, Hefei Normal University, Hefei 230061, China

**Keywords:** compressed sensing, underdetermined equation, spectral peak search, FFT

## Abstract

Spectral peak search is an essential part of the frequency domain parametric method. In this paper, a spectral peak search algorithm employing the principle of compressed sensing (CS) is proposed to rapidly estimate the spectral peaks. The algorithm adopts fast Fourier transform (FFT) with a few points to obtain the coarsely estimated spectral peak positions, and then only three small-scale inner products are iteratively calculated by increasing the input sequence length to rapidly refine the estimated positions. Compared with the conventional methods, this algorithm can directly capture the exact locations of spectral peaks without acquiring the entire spectrum. In addition, the proposed algorithm can be easily integrated into the existing frequency domain interpolation methods to accurately determine the spectral peak positions, and if so, only 30% of inner product operations of the original algorithms are required. Theoretical analysis and numerical results show that this algorithm yields accurate results with low complexity for analyzing both one-dimensional and two-dimensional signals.

## 1. Introduction

Estimation of the spectral peaks is an important work in frequency domain parametric estimators with various applications, such as the intelligent drive assistance system, the vital signs monitoring system, sonar, and radar [1,2,3,4]. Discrete Fourier transform (DFT) and its fast implementation, FFT, play a major role in estimating the spectral peaks. In many methods, FFT is first utilized to acquire the entire spectrum, and then the spectral peaks are determined by walking through each amplitude of the obtained spectrum. However, the picket fence effect of DFT and FFT, as is widely known, may lead to incorrect results. Up until now, the DFT-based interpolation methods have been proved to be an effective measure to solve this problem [5,6,7,8,9]. These methods usually consist of two steps: the coarse estimation stage and the fine estimation stage. In the coarse estimation stage, the methods commonly employ FFT to obtain the coarse frequencies, and in the fine estimation stage, the estimated frequencies are refined by interpolation. 

Most DFT-based interpolation methods have a low mean square error and signal-to-noise ratio (SNR) breakdown threshold; however, the computational complexities of the methods are higher than that of FFT due to their procedures including both FFT and inner products. In some applications, iteration or interpolation of more spectral lines is required to achieve more accurate results, which leads to more inner product operations. If the signal length is *N*, one more spectral line implies adding *N* multiplications. How to obtain the same accuracy as the existing frequency domain parametric method while significantly reducing the inner product operations is a research topic of significance. 

To achieve this, we employ the principle of compressed sensing (CS) and propose a new efficient spectral peak estimation algorithm based on an underdetermined equation. In recent years, many studies on the application of the CS technique in accurate recovery of the spectrum have been reported, e.g., [10,11,12]. Unlike these studies, our approach is focused on the spectral peak search, in which an underdetermined equation is firstly established based on the matrix form of DFT, and then an increasingly accurate spectral peak frequency can be estimated by a simple iteration using three small-scale inner products without accurately reconstructing the spectral coefficients.

## 2. CS-Based Spectral Peak Search Method

To improve the speed of the existing spectral peak estimation, this paper introduces the CS measurement principle into the DFT equations to obtain a super-fast spectral peak search model. In the new model, a few small-scale inner product operations take the place of the inner product operations in the DFT equations. The described model is shown in Figure 1.

### 2.1. The Proposed Spectral Peak Search Model

The inverse discrete Fourier transform (IDFT) equations can be represented as
(1)$$b=W_{N}^{-1}X,$$
where $W_{N}^{-1}$ ∈ ℂ*^N^*
^× *N*^ is the IDFT matrix. **b** ∈ ℂ*^N^*
^× 1^ is a time-domain column vector, and **X** is the discrete spectrum of **b**. To introduce the CS technology, the matrix equation is left-multiplied by a measurement matrix **Φ** ∈ ℂ*^n^*
^× *N*^ (*n* ≤ *N*): (2)$$\Phi b=\Phi W_{N}^{-1}X=W_{n}^{-1}X.$$

Viewing (2) as a CS problem has the following requirements. Firstly, the sensing matrix $W_{n}^{-1}$=$\Phi W_{N}^{-1}$ satisfies the restricted isometric property (RIP). Secondly, **X** can be sparsely represented.

Reference [13] has shown that the randomly selected partial Fourier matrix satisfies RIP. In this paper, the measurement matrix **Φ** is designed as a random partial unit matrix. Thus, $W_{n}^{-1}$=$\Phi W_{N}^{-1}$ is equivalent to the randomly selected partial Fourier matrix which satisfies RIP.

In the complete CS theoretical framework, **X** could be reconstructed by recovery algorithms along with a sparse matrix (denoted by **Ψ**) from solving
(3)$$\hat{\alpha }=\arg\min{\left\|\alpha \right\|}_{L}s.t.\Phi b=W_{n}^{-1}\Psi \alpha ,$$
(4)$$\hat{X}=\Psi \hat{\alpha }.$$

In which **α** is the sparse projection coefficients of **X**.

However, considering the measurement principle of the spectral peak search, there is no requirement for the reconstruction of **X**, as only the positions of the larger value in **X**, i.e., the spectral peak positions, need to be captured here. Thus, whether **X** is sparse has no essential effect on this method, and there is no need to introduce an additional sparse transformation even if **X** is not sparse. Then, we rewrite (2), and the following underdetermined equation can be constructed,
(5)$$b_{n}=W_{n}^{-1}X,$$

In which **b***_n_* ∈ ℂ*^n^* ^× 1^ is obtained by randomly extracting *n* rows from **b**, which is equivalent to **Φb** (as shown in (2)), and $W_{n}^{-1}$ ∈ ℂ*^n^* ^× N^ is acquired by the corresponding rows extraction from $W_{N}^{-1}$, i.e., partial Fourier matrix. In the view of CS theory, $W_{n}^{-1}$ can be regarded as the sensing matrix; consequently, **b***_n_* could be regarded as the corresponding measurement results.

Based on (5), the maximum position of **X** can be obtained by calculating the inner products between each column of $W_{n}^{-1}$ and **b***_n_*, which can be expressed as
(6)$$\max\left\{\langle W_{n1}^{-1},b_{n}\rangle ,\cdots ,\langle W_{nN}^{-1},b_{n}\rangle \right\},$$
in which $W_{n1}^{-1}$,…,$W_{nN}^{-1}$ represent the first column,…, and the *N*th column of $W_{n}^{-1}$. It is worth noting that despite the fact that (6) is used for the measurement of the single frequency signal, in the case of multi-frequency signal measurement, multiple peaks can also be determined from $\langle W_{n1}^{-1},b_{n}\rangle ,\cdots ,\langle W_{nN}^{-1},b_{n}\rangle$.

The above analysis is based on the IDFT, which changes the original inner product operations to the small-scale inner product operations. It can greatly reduce the computational complexity. Similarly, when the DFT-based interpolation techniques are introduced, which are equivalent to insert some columns in $W_{N}^{-1}$ (e.g., two DFT coefficients interpolation method [5,6], three DFT coefficients interpolation method [7,8]), the underdetermined equations can be also constructed by random extraction, and then the spectral peak positions are determined by small-scale inner product operations between each newly added columns in $W_{N}^{-1}$ and **b**.

### 2.2. The Proposed One-Dimensional Signal Processing Method 

As described in Section 2.1, the underdetermined equation model can achieve the spectral peak estimation by small-scale inner product operations. However, the direct application of this basic model for spectral peak search is still not as fast as FFT. Therefore, in Section 2.2 and Section 2.3, we present a coarse-to-fine strategy based on this model and discuss its specific steps for processing one-dimensional and two-dimensional signals, respectively.

In this section, we use the proposed model to estimate the spectral peaks of a one-dimensional signal. Taking the single-peak search as an example, the specific steps are as follows:

Coarse measurement:

The regular FFT is performed to a few previous discrete points of **b** ∈ ℂ*^N^* ^× 1^, denoted by $\hat{b}$∈ ℂ*^k^* ^× 1^ (*k < N*); thus, a coarse estimated spectral peak position is obtained;

2.Iterative operation:

The size of $\hat{b}$ is increased by adding a few subsequent points of **b** each time. Then, we can construct an underdetermined equation according to (5), i.e., *n* (*n* ≤ *k*) rows are randomly extracted from $\hat{b}$, and correspondingly *n* rows are extracted from $W_{N}^{-1}$. Then, we take the spectral peak position captured from the last iteration (for the first iteration, it is just the preliminary value obtained from the coarse measurement) as a prior knowledge to reduce the number of inner products. Specifically, supposing that the *j*th column of $W_{N}^{-1}$ (i.e., $W_{nj}^{-1}$) corresponds to the peak position at the last iteration, the comparison of inner products at the current iteration only needs to be implemented between **b***_n_* and $W_{n(j-1)}^{-1},W_{nj}^{-1},W_{n(j+1)}^{-1}$, which could be described as
(7)$$\max\left\{\langle W_{n(j-1)}^{-1},b_{n}\rangle ,\langle W_{nj}^{-1},b_{n}\rangle ,\langle W_{n(j+1)}^{-1},b_{n}\rangle \right\}.$$

The procedure of iterative operation is repeated until the end of **b**. As the length of $\hat{b}$ increases, the spectral peak position can continue to be updated by (7). The proposed one-dimensional signal processing method is shown in Algorithm 1.

**Algorithm 1:** The proposed algorithm for one-dimensional signal processing.**Input**: *A column vector of length N*: **b** ∈ ℂ*^N^* ^× 1^
*Number of sinusoid components: s*
*A column vector for coarse estimation:*
$\hat{b}$*∈* ℂ*^k^*^× 1^*, s < k < N*
*IDFT matrix:*
$W_{N}^{-1}$
*Points number added each time: δ*
*Extraction rate: β (β << 1)*
**Output**: *Spectral peak positions* **ẑ** = [*z*_1_,…, *z_s_*]^T^
**Initialize**: *Coarse spectral peak positions* **ẑ*_c_*** = zeros(*s*, 1)
1: /* *Coarse estimation stage* */
2: **Set**
$\hat{b}$ = **b**(1: *k*)
3: **Calculate**
*tempValue* = sort(FFT($\hat{b}$))
4: **Generate ẑ*_c_*** = *tempValue*(*k-s* + 1: *k*)
5:        /* *Iteration stage* */
6: **For**
*j from* 1 *to s* **THEN**
7:        **For**
*h from k* + 1 *to N*
**STEP**
*δ* **THEN**
8:        **Set**
$\hat{b}$(*h: h + δ* −1) = **b**(*h: h + δ*-1)
9:        **Calculate**
*n*= *β* × *h*
10:      **Obtain b***_n_ by randomly extracting n rows from*
$\hat{b}$
11:      **Obtain**
$W_{n}^{-1}$ *by correspondingly extracting n rows of*
$W_{N}^{-1}$
12:      **IF**
*h* == *k* + 1 **THEN**
13:                       **Obtain**
$z=\max\left\{\langle W_{n({\hat{z}}_{c}(j)-1)}^{-1},b_{n}\rangle ,\langle W_{n({\hat{z}}_{c}(j))}^{-1},b_{n}\rangle ,\langle W_{n({\hat{z}}_{c}(j)+1)}^{-1},b_{n}\rangle \right\}$
14:      **Else**

15:                       **Obtain**
$z=\max\left\{\langle W_{n(z-1)}^{-1},b_{n}\rangle ,\langle W_{n(z)}^{-1},b_{n}\rangle ,\langle W_{n(z+1)}^{-1},b_{n}\rangle \right\}$
16:      **End IF**
17:      **END FOR**
18:      **Set**
*z_j_* = *z*
19: **END FOR**
20: **Return**
**ẑ**


### 2.3. The Proposed Two-Dimensional Signal Processing Method 

The size of the signal is assumed as *N × M*, denoted by **B** = {**b**_1_, **b**_2_,…, **b***_M_*}, and **b***_i_* (*i* = 1, 2,…, *M*) could be regarded as a one-dimensional signal. In this section, we only discuss the case in which a column vector consisting of the spectral peak value of each **b***_i_* is regarded as the input of the second dimension transformation.

Coarse measurement:

We select a few previous columns of **B**, denoted by $\hat{B}$ = {**b**_1_, **b**_2_,…,**b***_k_*} (*k < M*). Then, the spectral peak of each column of $\hat{B}$ can be determined by regular FFT, and this procedure can be expressed as
(8)$$b_{i}^{\max}=\max(\mathrm{FFT}(b_{i}))\left(i=1,2,\cdots ,k\right),$$

In which $b_{i}^{\max}$ is the peak value of spectrum of **b***_i_*. Thus, a column vector composed of the peak value of each spectrum can be defined as ${\hat{B}}_{x}$ = [$b_{1}^{\max}$, $b_{2}^{\max}$,⋯, $b_{x}^{\max}$]^T^ (k < x < M), which is the input data of the second-dimension transformation. Continuously, the regular FFT is performed to ${\hat{B}}_{x}$ in order to obtain the coarse spectral peak positions, denoted by **Ẑ***_c_* = [*Ẑ*_1_, *Ẑ*_2_, ⋯, *Ẑ_s_*]^T^, in which *s* represents the number of spectral peaks;

2.Iterative operation:

As a few subsequent columns of **B** are joined to $\hat{B}$, the peak value of each spectrum is added to ${\hat{B}}_{x}$, correspondingly. Similarly to the iterative operation in Section 2.2, we can take the last spectral peak position of ${\hat{B}}_{x}$ as a priori knowledge. With the increasing length of ${\hat{B}}_{x}$, the spectral peak position of newly ${\hat{B}}_{x}$ is continuously obtained by three inner products. The proposed two-dimensional signal processing method is shown in Algorithm 2.

**Algorithm 2:** The proposed method for two-dimensional signal processing.**Input**: *A two-dimensional signal*: **B***∈* ℂ*^N^* ^× *M*^
*Number of estimated spectral peaks: s*
*A matrix for coarse estimation:*
$\hat{B}$*∈* ℂ*^N^* ^× *k*^*, s < k < M*
*Input column vector of the second dimensional transformation:*
${\hat{B}}_{x}$
*Columns number added each time: δ*
*IDFT matrix:*
$W_{N}^{-1}$
*Extraction rate: β (β<<1)*
**Output**: *Spectral peak positions:* **Ẑ** = [*Z*_1_,…, *Z_s_*]^T^
**Initialize**: *Coarse spectral peak positions:* **Ẑ***_c_* = zeros (*s*, 1)
1:          /* *Coarse estimation stage* */
2: **Set**
$\hat{B}$ = **B** (:, 1: *k*)
3: **For**
*i* from 1 to *k* **THEN**
4:          **Calculate**
$b_{i}^{\max}$ = max (FFT($\hat{B}$(:, *i*)))
5:          **Obtain**
${\hat{B}}_{x}$(j) = $b_{i}^{\max}$
6: **END FOR**
6: **Perform**
*tempValue* = sort (FFT(${\hat{B}}_{x}$))
6: **Generate Ẑ***_c_* = *tempValue* ((*k*-*s*): *k*)
5:          /* *Iteration stage* */
6: **For**
*j from* 1 *to s* **THEN**
7:          **For**
*h from* (*k* + 1) *to M* **STEP**
*δ* **THEN**
8:          **Generate**
$\hat{B}$(:, *h*: *h + δ*-1) = **B** (:, *h*: *h + δ*-1)
3:                       **For**
*i* from *h* to (*h + δ*-1) **THEN**
4:                                          **Calculate**
$b_{i}^{\max}$ = max (FFT($\hat{B}$(:, *i*)))
4:                                          **Set**
${\hat{B}}_{x}$(*i*) = $b_{i}^{\max}$
5:                       **END FOR**
9:          **Calculate**
*n = β × h*

10:        **Obtain b***_n_ by randomly extracting n rows from* ${\hat{B}}_{x}$
11:        **Obtain**
$W_{n}^{-1}$ *by correspondingly extracting n rows of*
$W_{N}^{-1}$
12:        **IF**
*h == k* + 1 **THEN**
13:                                **Obtain** $z=\max\left\{\langle W_{n({\hat{z}}_{c}(j)-1)}^{-1},b_{n}\rangle ,\langle W_{n({\hat{z}}_{c}(j))}^{-1},b_{n}\rangle ,\langle W_{n({\hat{z}}_{c}(j)+1)}^{-1},b_{n}\rangle \right\}$
14:        **Else**
15:                                  **Obtain**
$z=\max\left\{\langle W_{n(z-1)}^{-1},b_{n}\rangle ,\langle W_{n(z)}^{-1},b_{n}\rangle ,\langle W_{n(z+1)}^{-1},b_{n}\rangle \right\}$
16:        **End IF**
17:        **END FOR**
18:        **Set**
*z_j_* = *z*
19: **END FOR**
20: **Return**
**ẑ**


### 2.4. Complexity Analysis

In this section, we analyze the multiplicative complexity of both one-dimensional signal processing and two-dimensional signal processing using the proposed method.

For one-dimensional signal processing, the multiplication complexity is determined by both the coarse estimate stage and the iteration stage. In the coarse estimate stage, the multiplication complexity can be expressed as $O\left(k\log_{2}^{k}\right)$. In the iteration procedure (one can assume the total sequence length during *h*th iteration is *p*), the multiplication complexity during one iteration procedure is *O*(*s*3*βp*). Thus, the total multiplication complexity of each output is
(9)$$O(k\log_{2}^{k}+s3\beta p),$$
and the corresponding multiplication complexity of the *N*-point FFT algorithm is $O\left(N\log_{2}^{N}\right)$. To clearly see the computational advantage of the proposed algorithm, we set the following typical parameters: *N* = 512, *k* = 32, *β* = 30%, *s* = 2, *p* = 180, and the ratio of the computational complexity of two algorithms is
(10)$$\frac{O\left(N\log_{2}^{N}\right)}{O\left(k\log_{2}^{k}+s3\beta p\right)}=9.52.$$

For two-dimensional signal processing, the multiplication complexity of the coarse estimation stage is composed of *k* times *N*-point FFT and one time *k*-point FFT. In the iteration procedure, it is assumed that the total column number during the *h*th iteration is *p*, and then the multiplication complexity of this iteration procedure consists of *δ* times *N*-point FFT and three small-scale inner product operations. Thus, the total multiplication complexity of each output is
(11)$$O\left(kN\log_{2}^{N}+k\log_{2}^{k}+\delta N\log_{2}^{N}+s3\beta p\right),$$

Additionally, the corresponding multiplication complexity of the 2-D FFT algorithm is $O\left(MN\log_{2}^{N}+M\log_{2}^{M}\right)$. We also set the following typical parameters: *N* = 512, *M* = 512, *δ* = 2, *k* = 32, *β* = 30%, *s* = 2, *p* = 180, and the ratio of the computational complexity of the two algorithms is
(12)$$\frac{O\left(MN\log_{2}^{N}+M\log_{2}^{M}\right)}{O\left(kN\log_{2}^{N}+k\log_{2}^{k}+\delta N\log_{2}^{N}+s3\beta p\right)}=15.04.$$

The main advantage of the proposed method is that it greatly reduces the scale of inner product operations during the spectral peak search (described in Section 2.1). Since the inner product operations will inevitably exist in the traditional DFT and its interpolation techniques, the proposed method can be applied to improve DFT and its interpolation techniques. Compared with the original methods, the proposed algorithm has the same accuracy with lower computational complexity.

### 2.5. Application of the Proposed Method

The spectral peak estimation of one-dimensional and two-dimensional signals is a much-studied problem with many practical applications, such as machine fault diagnosis system (one-dimensional signal), Doppler radar system (one-dimensional signal), sonar detection system (one-dimensional signal), and sawtooth linear frequency modulated continuous wave (LFMCW) system (two-dimensional signal). This paper takes the sawtooth LFMCW radar system as an example. The beat signal of LFMCW radar contains both distance information and Doppler information (i.e., speed) of the moving target, and 2-D FFT is usually used to obtain the distance and speed parameters. The beat signal can be expressed as [14]
(13)$$S_{\mathrm{IF}}(t)=e^{j\left\{2\pi \left(f_{R}t+f_{D}t^{2}\right)+\phi \right\}},$$
where **S**_IF_(*t*) is the beat signal, *f*_R_ is the frequency of **S**_IF_(*t*), and *f*_D_ is the Doppler shift caused by the moving target. According to [14], *f*_R_ can be determined by
(14)$$f_{R}=K\frac{2R_{0}}{c}+\frac{2V}{\lambda }.$$
in which *K* is the frequency slope constant, *R*_0_ is the initial distance, c is the speed of light, *λ* is the wavelength, *V* is the target speed, and *f*_D_ can be obtained form
(15)$$f_{D}=\frac{2KV}{c}.$$

It can be clearly seen from (14) and (15) that we can obtain real-time distance and speed by estimating *f*_R_ and *f*_D_, respectively. Furthermore, *f*_R_ is estimated by performing first-dimensional FFT, and *f*_D_ can be estimated by performing second-dimensional FFT. Therefore, the method described in Section 2.3 can be used for the LFMCW radar system.

## 3. Simulation Results and Discussion

Two sets of numerical simulations are provided to examine the performance of the proposed method. The first group examines the effectiveness of the suggested method for one-dimensional signal processing. The second group examines the effectiveness of the suggested method for two-dimensional signal processing. A total of 100 runs for each experiment are conducted.

### 3.1. One-Dimensional Signal Processing

The first group includes four parts. First, to obtain a valid value of extraction rate, i.e., *β*, we gradually increase *β* from less to more, and calculate the relative frequency errors. Second, the spectrum of the proposed method is compared with the FFT spectrum. Third, we integrate the proposed method into the existing frequency interpolation method and compare their results. Finally, the noise performance of the proposed algorithm is presented. The experimental parameters are set as follows: the input sequences are obtained from *sin*(2 × pi × 207.7 × [0: 511]), and the sampling frequency is 1000 Hz, *k* = 32, *δ* = 2, and SNR = 3. The relative position error is defined as
(16)$$E_{r}=\frac{\left|f_{0}-f_{1}\right|}{f_{0}}\times 100\%$$
in which |∙| means absolute value. *E*_r_ is the relative frequency error. *f*_1_ is the frequency obtained by one iteration and *f*_0_ is the reference value. In Figure 2, the reference value is frequency calculated by 512-point FFT, i.e., the vertical axis in Figure 2.

As it can be seen from Figure 2, the valid extraction rates are associated with the sequence length during one iteration, and the higher *p* is, the lower *β* will be.

Then, we show the spectrum of one iterative procedure (based on Figure 2, we choose *p* = 180, *β* = 20%) and compare it with the 512-point FFT spectrum.

It can be most clearly seen from Figure 3 that the spectrum amplitudes of the proposed method are smaller than that of the 512-point FFT; however, the peak position of the two methods are the same. In practice, due to the picket fence effect associated with the 512-point FFT, the spectral peak position cannot be obtained as decimals in those methods, only as whole numbers. Therefore, the peak frequencies obtained by the two methods are different from the true frequency. It is worth noting that many small-scale inner products have been calculated in Figure 3 to intuitively display the spectrum of the proposed method, but in fact, only three small-scale inner products are required.

Third, we do not change the calculation procedures of the method in [9], and only use the proposed method to significantly reduce the inner product operations of Mou’s method. The experimental parameters for the integrated method are set as follows, *β* = 20%, *k* = 32, and *δ* = 2. In Figure 4, the reference value is the true frequency, i.e., 207.7 Hz.

The proposed method, for which *p* is more than 100, will obtain the same accuracy as the method in [9], as is shown in Figure 4.

To demonstrate the noise immunity of the proposed method, we calculated the required *β* for each SNR under the condition of *E*_r_ < 1% (the reference value is frequency calculated by 512-point FFT).

We present the relationship between *β*, SNR, and *p* under the set condition, as shown in Figure 5. The four curves represent four iterative procedures with different sequence lengths, from which we can draw the following conclusions: the higher *p* is, the lower *β* will be; the higher SNR is, the lower *β* will be.

### 3.2. Two-Dimensional Signal Processing

This experiment takes the beat signal of the sawtooth LFMCW radar system as an example. The beat signal contains both distance and speed information of the targets, and 2-D FFT is the most common method for processing the beat signal. It is assumed that a 512 × 512 matrix is obtained by sampling the beat signal, of which each column is a chirp signal, and each chirp signal contains 512 elements. There are two objects with speeds of 3.5 m/s and 4.9 m/s and initial distances of 12 m and 10.7 m, respectively. The other experimental parameters are set as follows: *k* = 32, *δ* = 2, *p* = 180, *β* = 20%, *s* = 2, and SNR = 3.

Figure 6 presents the same conclusions as those obtained from the one-dimensional signal processing, i.e., the spectral peak positions of two methods are the same and the spectral lines have different amplitudes. Here, to clearly describe the differences between two partial spectrums, many small-scale inner products have also been calculated, but in practice, only three small-scale inner products are required since the range of the spectral peak position has established a priori knowledge from the last iteration.

## 4. Conclusions

In this study, the spectral peak positions of both one-dimensional signal and two-dimensional signals were rapidly obtained by a simple iterative procedure based on the underdetermined equation. During the course of an iteration, a few points were added and only three small-scale inner product operations were performed. In addition, this algorithm can be easily applied to other frequency interpolation methods, and the integrated spectral peak estimation algorithm can obtain the same spectral peak positions as the original method, but with only 30% of the inner product operations. In summary, the proposed algorithm is suitable for real-time operating systems as well as accurate measurements systems. We will consider introducing some key techniques of machine learning, e.g., atomic functions [15], to improve the proposed model in further studies.

## Figures and Tables

**Figure 1 sensors-22-07025-f001:**
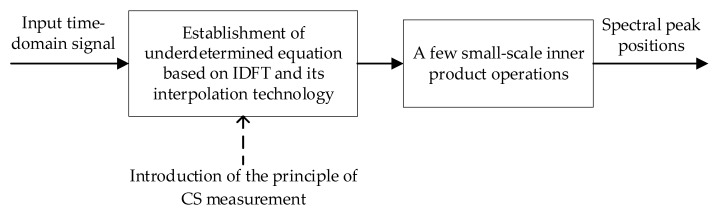
Basic model for super-fast spectral peak search.

**Figure 2 sensors-22-07025-f002:**
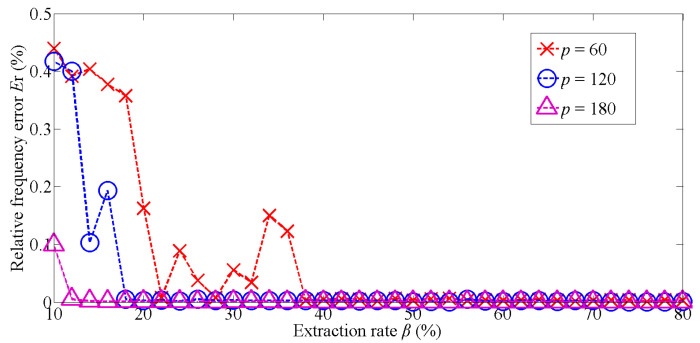
Relationship between *β* and *E*_r_ during one iteration with different sequence lengths (i.e., *p*).

**Figure 3 sensors-22-07025-f003:**
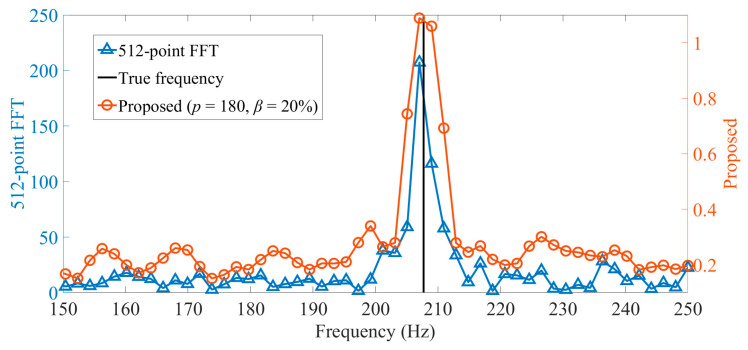
Two partial spectrums (the frequency ranges are from 150 Hz to 250 Hz) are obtained by the proposed method during one iteration and 512-point FFT, respectively.

**Figure 4 sensors-22-07025-f004:**
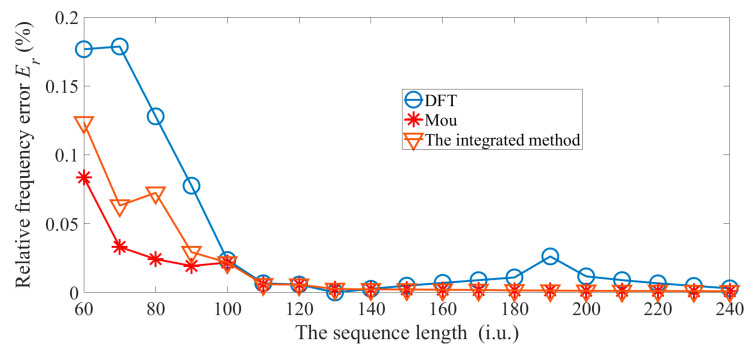
Relative frequency errors are obtained by increasing the sequence length.

**Figure 5 sensors-22-07025-f005:**
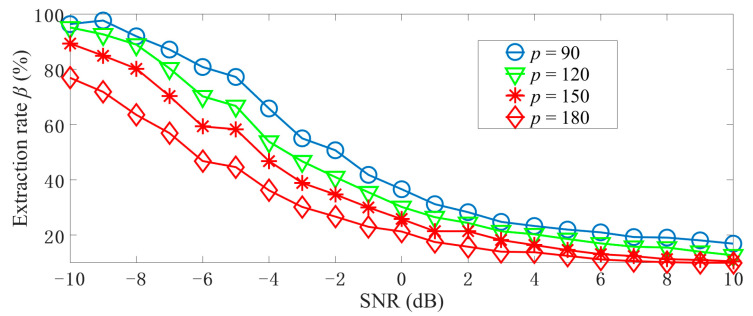
The noise performance of the proposed method during the iterative.

**Figure 6 sensors-22-07025-f006:**
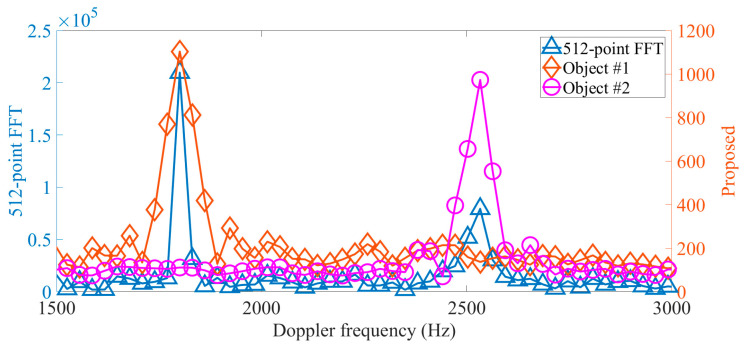
Two partial Doppler spectrums are obtained by the proposed method during one iteration and 512-point FFT, respectively.

## Data Availability

Not applicable.

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
