# Peer review of "An Efficient CS-Based Spectral Peak Search Method"

_sensors, 2022, doi:10.3390/s22187025_

Round 1

Reviewer 1 Report

The paper offers an interesting algorithm of finding peaks with advanced matrix calculations leading to faster conversions which may be useful for super-fast signals analysis.  

While I support the paper, can I suggest to include the Picture at the top which simply demonstrate the problem and can be visually understood by readers?

Second suggestion is for authors to explore the theory of Atomic Functions and Atomic Machine Learning techniques (18) (PDF) Atomic Machine Learning (researchgate.net). Finite Atomic Functions can exactly reproduce sections of polynomials and smooth analytic functions which may lead to increased efficiency, 

Author Response

Thank you very much for all comments which are very valuable and meaningful to improve the quality of this manuscript. We have carefully revised the manuscript according to all the suggestions from reviewer, and replied to the comments (see attachment).

Thanks.

Reviewer 2 Report

1.       The authors should justify the main advantages of the present study in comparison with the existing studies.

2.      Some grammatical errors have been noticed throughout the manuscript. I hope the authors can revise the whole manuscript

Author Response

(The authors gave the same response as above.)
